# Study on Mechanical Properties of Ring Sandstone Specimen under Temperature and Water Coupling Dynamic Compression

**Qi Ping** [1,2,3], **Qi Gao** [2,3,*] **and Shiwei Wu** [2,3]

1 State Key Laboratory of Mining Response and Disaster Prevention and Control in Deep Coal Mine, Anhui University of Science and Technology, Huainan 232001, China
2 Engineering Research Center of Mine Underground Projects, Ministry of Education, Anhui University of Science and Technology, Huainan 232001, China
3 School of Civil Engineering and Architecture, Anhui University of Science and Technology, Huainan 232001, China
* Correspondence: 2020200277@aust.edu.cn; Tel.: +86-17355450517

**Abstract:** The excavation of hard rock roadways in coal mines is often in the environment of underground water and high ground temperature, and it is easy to be affected by the dynamic load, which leads to roadway destruction and increases the difficulty of roadway support. The ring sandstone specimens with different inner diameters (0~25 mm) were treated with temperature and water coupled, and the dynamic compression test was produced by the Hopkinson pressure rod device (SHPB). The experimental results indicate that the coupling effect of temperature and water reduces the dynamic performance of sandstone specimens. XRD test results showed that the composition of sandstone specimens did not change before and after warm water coupling, and no new substances were found. Dynamic properties of ring sandstone specimens with different inner diameters weaken with the increase in inner diameters. With the increasing inner diameter of ring sandstone specimens, the energy dissipation per unit volume increases the dynamic compressive strength decreases, and the degree of breakage increases. Fracture morphology, average strain rate, and dynamic peak strain of ring sandstone specimens increase with inner diameter.

**Keywords:** rock dynamics; temperature and water coupled; ring specimen; dynamic compression; SHPB





## 1. Introduction

Coal mine roadways created during tunneling are complex temperature, fluid saturation, and loading environments, posing great challenges in roadway excavation and support [1]. In the process of mining, rock burst is easy to cause natural disasters such as rock bursts, which affects construction safety [2,3]. Studying the dynamic characteristics of roadway rocks has great engineering significance under dynamic load and provides some data support for mine roadway support and excavation.

Scholars related to the field of rock have made fruitful achievements in rock statics. Ping et al. [4] analyzed the dynamic properties of sandstone specimens after various temperature cycles, and the results displayed that the dynamic properties changed with the difference in temperature. Yang et al. [5] carried out cyclic loading of different water-bearing sandstones used digital speckle technology to reveal the crack growth and failure law of sandstone specimens, and observed gradual change from tensile failure to shear failure in specimens. Xu et al. [6] considered the hysteresis of rock burst, used rock samples to conduct time-delay uniaxial compression tests, and found that the stress level had a significant impact on the time-delay failure of rock samples. Jia et al. [7] carried out high-temperature treatment on three different types of rock samples, followed by water cooling and cyclic loading, and found that the three rock samples showed a decrease in plastic deformation, an increase in elastic modulus and an increase in uniaxial compressive

strength as the number of cycles increased. Ping et al. [8] analyzed the dynamic splitting mechanical properties of sandstone specimens after treatment with temperature and water and found that with an increase in loading rate, the specimens' dynamic tensile strength increased due to strain rate effects. Deng et al. [9] used two water-bearing rock specimens to study dynamic mechanical and found that water content affected fracture mechanical and acoustic emission characteristics of rock samples. Ai et al. [10] conducted dynamic mechanical properties tests on coal rock with different bedding and found that crack growth path and dynamic properties of coal samples were affected by layering direction. Ping et al. [11] found that the dynamic tensile strength of limestone increases with absorbed energy by using the static and static combination. Sun et al. [12] used a $MgSO_4$ solution to process the dry and wet cycle of sandstone specimens, and found that the dry and wet cycle of salt solution had a greater impact than that only soaked in water. He et al. [13] conducted freeze-thaw cycles on sandstone specimens with various water-bearing states and put forward that the peak shear stress decreased with more cycles. Wang et al. [14] conducted an experimental study about rock samples at different temperatures and with different water content; it was found that water-bearing rock samples lose tensile strength as water content increases. Wang et al. [15] combined laboratory tests with numerical simulation and found that the permeability of red sandstone changed little at the initial and elastic deformation stages but changed faster when it reached the peak stress. Zhao et al. [16] conducted hydraulic coupling tests on fractured limestone and verified the validity conditions of the Mohr Coulomb principle. Chen et al. [17] conducted nuclear magnetic resonance and shear creep tests on freeze-thawing red sandstone and found that water content increased the creep variable in red sandstone. Yang et al. [18] used red sandstone, after freezing at low temperatures, found that the low-temperature environment caused the first damage of the cementing material inside the red sandstone, resulting in the fracture of the specimen. Xue et al. [19] carried out cyclic impact loading on the white sandstone after the coupling action of chemical corrosion and axial load and found that when Ph = 7 and axial pressure is 12.6 Mpa, the impact failure strength of the white sandstone reaches the maximum. Ping et al. [20] studied the progressive compressive strength of sandstone specimens after alkali chemical corrosion and found that the dynamic tensile strength and elastic modulus of sandstone specimens decreased significantly after 28 days of corrosion. Li et al. [21] conducted cooling treatment on the heated sandstone at different cooling rates and found that the higher the cooling rate, the more serious the damage to the specimen. Fan et al. [22] tested the Schmidt hardness of drying and different water absorption time points by using an L-shaped Schmidt hammer and found that the Schmidt hardness of red sandstone decreased with the increase in water absorption time and water absorption rate. Laubach et al. [23] conducted geophysical research from a chemical perspective in order to correctly identify and interpret cracks in geophysical measurements during field characterization and monitoring of underground engineering activities. Ping et al. [24] conducted dynamic compression mechanical properties tests on prefabricated fractured sandstone specimens with different angles and found that when the dip Angle was 45°, the fracture morphology of specimens was the largest. Zhao et al. [25] established a new strength model of natural rocks and proposed a parameter that can more fully reflect the influence of joint topography parameters on shear strength.

Under dynamic loading, coal mine roadway rock was studied in order to understand its dynamic behavior. Many scholars use ring rock specimens to analyze various mechanical property parameters of roadway rocks. Yang et al. [26] studied the mechanical compression properties of the ring specimen through numerical simulation and found that the friction coefficient could be used as the intrinsic property of the specimen. Ping et al. [27] used eight-ring sandstone specimens after different temperature-water coupling treatments. When the water temperature was 45 °C, ring sandstone specimens achieved their maximum dynamic tensile strength. Wang et al. [28] conducted a temperature-wet cycle treatment on the ring granite and then carried out a radial compression mechanical property test. According to the study, the ring specimen cracked first along its inner wall, and then the crack spread to

the outer wall of the specimen, resulting in its instability and failure. Wu et al. [29] studied the stress characteristics at the hole wall of ring sandstone specimens with different inner diameters and found that the peak load of ring specimens decreased with the increase in aperture. Huang et al. [30] used ring granite specimens to analyze the physical and mechanical properties by different temperatures and found the tensile strength increased first and then decreased. Li et al. [31] researched the static and dynamic splitting mechanical properties with virous inner diameters and found that the dynamic tensile strength of ring specimens was about 5 times the static tensile strength. Yang et al. [32] found that the radius ratio and eccentricity had an impact on the deformation and strength of the disk. Zhao et al. [33] conducted triaxial tests on channel sandstone under various confining and pore pressures and found when specimens were coupled to water and subjected to deformation, their peak deviatoric stress strength and deformation modulus were generally lower than when they were isolated from water. Wang et al. [34] found that the specimen's failure modes were mainly split failure and shear failure by using precast cavity rock-like concrete materials.

It can be seen from these research results that progress has been made in the study of rock statics and dynamics under complex environmental conditions. Based on the innovations of scholars, this paper considers that the roadway rocks of coal mines are vulnerable to the influence of groundwater and high ground temperature, and dynamic loads often occur during excavation and tunneling, which makes roadway support more difficult. Therefore, the ring sandstone specimens with different inner diameters (0~50 mm) were used to simulate the different wall thicknesses of roadway rocks and were coupled with warm water. (Split Hopkinson Pressure Bar, SHPB) was conducted to study the dynamic compression mechanical properties of the treated ring sandstone specimen, and its dynamic performance parameters were analyzed to provide a reference for coal mine roadway support.

## 2. Specimen Processing and Basic Physical Parameter Change

### 2.1. Preparation of Specimen

Testing was conducted on rock samples taken from the Ding-ji Coal Mine, Huainan City, Anhui Province. According to the international rock mechanics test code and the Chinese rock dynamics test code [35,36], the rock samples were cored, cut, and polished into standard cylindrical specimens with a diameter of D = 50 mm and H = 25 mm. Drill bits *d* = 5, 10, 15, 20, and 25 mm with different outer diameters were used to drill sandstone specimens. The processed ring sandstone specimen is shown in Figure 1.

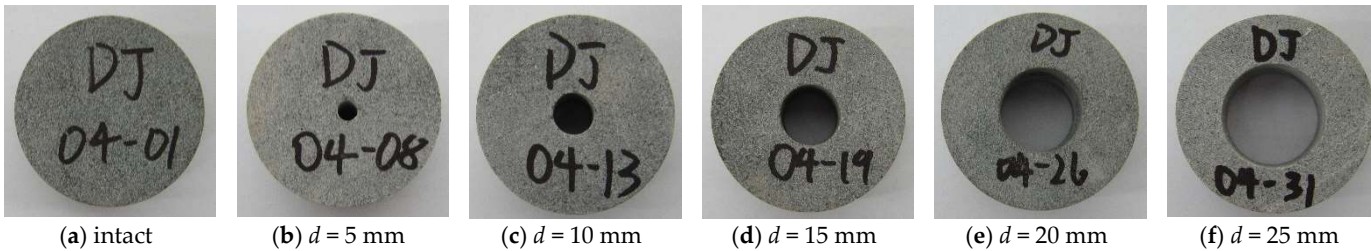

| (**a**) intact | (**b**) *d* = 5 mm | (**c**) *d* = 10 mm | (**d**) *d* = 15 mm | (**e**) *d* = 20 mm | (**f**) *d* = 25 mm |

**Figure 1.** Processed annular sandstone specimen.

### 2.2. Specimen Coupling Treatment with Temperature and Water

The prepared ring sandstone specimen was put into a constant temperature water bath and soaked for 48 h so that the specimen reached the saturated state. In the study of Ping et al. [37], 45 °C is an inflection point at which the dynamic properties of specimens change. Therefore, 45 °C was adopted as the coupling saturation condition of warm water in this experiment. The thermostatic water bath used in the test is shown in Figure 2.

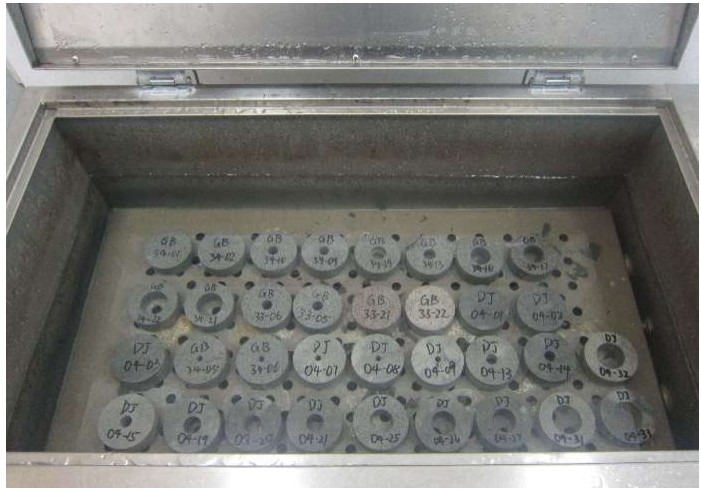

**Figure 2.** Water bath box.

The ring sandstone specimen was coupled with temperature and water, the surface color of the specimen was slightly lightened, and there were small particles spalling on the surface.

### 2.3. Basic Physical Parameters of the Specimen

In Table 1, the physical parameters of specimens before and after temperature and water coupling have been provided.

**Table 1.** Comparison of physical data before and after temperature-water coupling.

| Inner Diameter (mm) | Specimen Number | Before Temperature and Water Coupling | | | After Temperature and Water Coupling | | |
|---|---|---|---|---|---|---|---|
| | | Mass (g) | Volume (mm$^3$) | Density (g/mm$^3$) | Mass (g) | Volume (mm$^3$) | Density (g/mm$^3$) |
| 0 (Intact) | DJ04-01 | 125.92 | 49.06 | 2.57 | 125.95 | 49.53 | 2.54 |
| | DJ04-02 | 125.39 | 49.02 | 2.56 | 125.42 | 49.50 | 2.53 |
| | DJ04-03 | 126.92 | 49.27 | 2.58 | 126.95 | 49.73 | 2.55 |
| 5 | DJ04-07 | 125.39 | 48.56 | 2.58 | 125.55 | 48.95 | 2.57 |
| | DJ04-08 | 124.39 | 48.55 | 2.56 | 124.60 | 48.91 | 2.55 |
| | DJ04-09 | 126.36 | 48.56 | 2.60 | 126.38 | 48.95 | 2.58 |
| 10 | DJ04-13 | 121.69 | 47.31 | 2.57 | 121.86 | 47.62 | 2.56 |
| | DJ04-14 | 118.25 | 46.99 | 2.52 | 118.52 | 47.32 | 2.50 |
| | DJ04-15 | 121.56 | 47.17 | 2.58 | 121.81 | 47.48 | 2.57 |
| 15 | DJ04-19 | 114.55 | 44.75 | 2.56 | 114.77 | 44.98 | 2.55 |
| | DJ04-20 | 114.59 | 44.66 | 2.57 | 114.88 | 44.88 | 2.56 |
| | DJ04-21 | 114.00 | 44.76 | 2.55 | 114.35 | 44.94 | 2.54 |
| 20 | DJ04-25 | 103.42 | 41.23 | 2.51 | 103.67 | 41.42 | 2.50 |
| | DJ04-26 | 103.07 | 41.03 | 2.51 | 103.36 | 41.20 | 2.51 |
| | DJ04-27 | 100.25 | 40.96 | 2.45 | 100.51 | 41.11 | 2.45 |
| 25 | DJ04-31 | 92.36 | 36.72 | 2.52 | 92.64 | 36.85 | 2.51 |
| | DJ04-32 | 90.98 | 36.77 | 2.47 | 91.24 | 36.89 | 2.47 |
| | DJ04-33 | 91.17 | 36.83 | 2.48 | 91.42 | 36.95 | 2.47 |

Both the mass and volume increased to a certain extent, but the density decreased. In addition to calculating the mass, volume, and density growth rate of the specimen, a preliminary analysis is conducted of its basic physical properties.

The mass growth rate is shown in Figure 3.

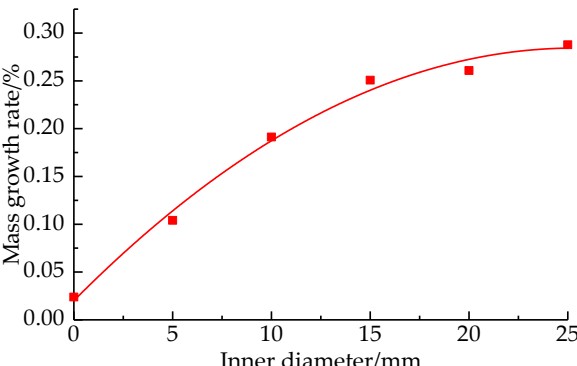

**Figure 3.** Mass growth rate changes with annular diameter change.

The mass growth rate of ring sandstone specimens is related to specimen diameter by a quadratic function, and the positive correlation is obvious. After data fitting, the correlation coefficient reached 0.9879, showing a strong positive correlation. The fitting formula is shown in (1).

$$m\prime = -4.108 \times 10^{-4}d^2 + 0.021d + 0.020 \ (R^2 = 0.9879) \tag{1}$$

where: $m'$ is the mass growth rate.

Analysis of the reasons, as the inner diameter of the ring sandstone specimen increases, the surface of the surface between the inner wall of the specimen and the water is increased, and the water is more easily admitted into the interior of the specimen. The mass of the ring sandstone specimens increases due to water infiltration of internal cracks. The mass growth rate of ring sandstone specimens increases with the increase in inner diameter.

Figure 4 provides the variation trend of volume growth.

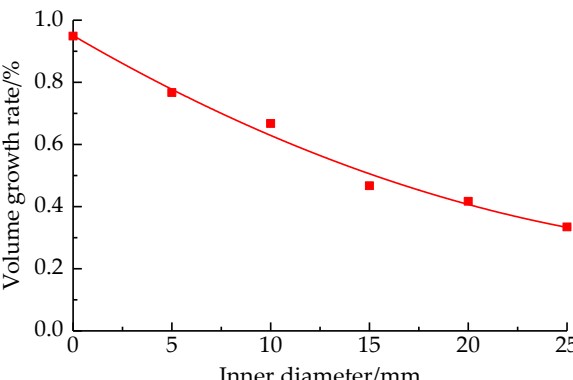

**Figure 4.** Volume growth rate changes with annular diameter change.

The volume growth rate of ring sandstone specimens decreases with the increase in inner diameter. The fitting results show a quadratic function where the inner diameter of the ring specimen is with the volume growth rate, and the correlation coefficient reaches 0.9808. The fitting formula is shown in (2).

$$V\prime = 4.961 \times 10^{-4}d^2 - 0.037d + 0.950 \ \ (R^2 = 0.9808) \tag{2}$$

where: $V'$ is the volume growth rate.

Analysis of the reason: the ring sandstone specimen contains holes and micro-cracks. The cracks in sandstone specimens are easy to expand after the coupling action of temperature and water. Therefore, the volume tends to increase. By observing the surface of the sandstone specimen after the temperature and water coupling, it is found that there are fine particles spalling off on the surface, and the specimen deteriorates, damaged, and the

volume decreases to a certain extent. Therefore, as the inner diameter of the ring sandstone specimen increases, the volume growth rate shows a decreasing trend.

Figure 5 showed the relationship between the density reduction rate and the inner diameter.

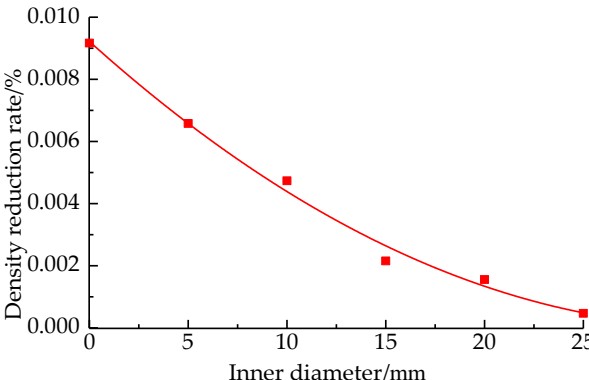

**Figure 5.** The density reduction rate varies with annular diameter change.

After data fitting, it is found that the density reduction rate of ring sandstone specimens shows a quadratic function decreasing relationship with the change of inner diameter, and the correlation coefficient reaches 0.9879. The fitting formula is shown in (3).

$$\rho\prime = 8.013 \times 10^{-6}d^2 - 5.721 \times 10^{-4}d + 0.009 \quad (R^2 = 0.9879) \tag{3}$$

where: $\rho'$ is the density growth rate.

Analysis reason: Influenced by the coupling effect of temperature and water, the mass and volume of ring sandstone specimens have an increasing trend. The volume growth rate is slightly higher than the mass growth rate, so the density of the specimen will decrease.

### 2.4. Microscopic Material Composition

The composition of a specimen before and after temperature and water coupling was detected by (X-ray Diffraction, XRD) method. The consequence is shown in Figure 6.

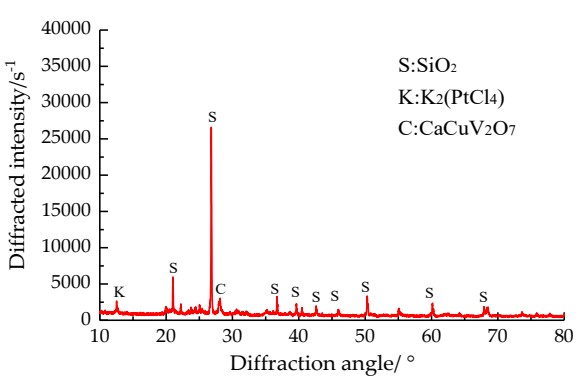

(**a**) Before Temperature and Water Coupling

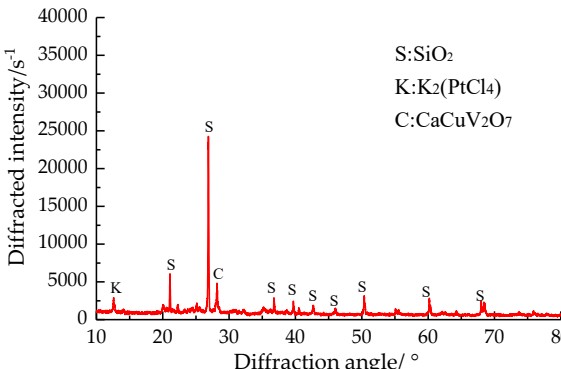

(**b**) After Temperature and Water Coupling

**Figure 6.** XRD pattern of sandstone specimen.

As can be found in Figure 6, no new substances were found before and after the temperature and water coupling. The composition of ring sandstone specimens remains basically unchanged before and after temperature and water coupling, and quartz (SiO2) is one of the most important components of sandstone specimens.

## 3. Specimen Processing and Basic Physical Parameter Change

### 3.1. SHPB Test Facility

By using the Hopkinson pressure bar (SHPB) test facility, dynamic compression mechanical properties of ring sandstone specimens with different inner diameters were studied could be tested. This test equipment was the SHPB test facility of the State Key Laboratory of Mining and Response in the Deep Coal Mine. The pressing rod of the equipment is made of high-strength alloy steel, with a diameter = 50 mm, length of the incident rod = 2 m, length of the transmission rod = 1.5 m, wave velocity of the press rod = 5380 m/s, density = 7636 kg/m$^3$, Poisson's ratio = 0.28. The test equipment is shown in Figure 7.

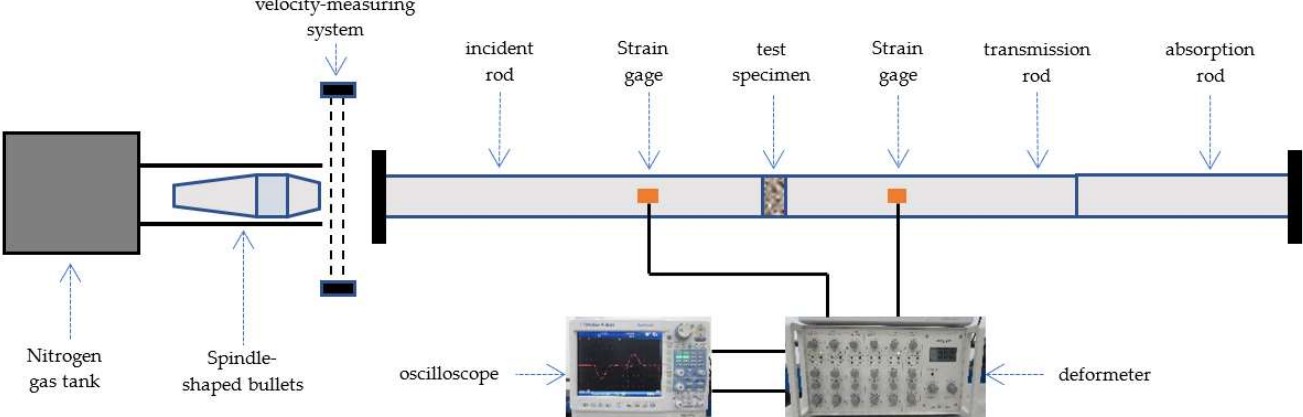

**Figure 7.** SHPB test device.

SHPB equipment is mainly composed of a nitrogen loading system, bullet, velocity measurement system, oscilloscope and strain gauge, and other components. The impact pressure of 0.4 MPa was used to load the ring sandstone specimen under action load.

The two-wave formula was used to process the data collected by the strain gauge, and various dynamic parameters, axial load $P(t)$, peak strain $\varepsilon(t)$, and average strain rate $\dot{\varepsilon}(t)$ of the ring sandstone specimen were obtained. The formula is shown in (4).

$$\left.\begin{array}{l} P(t) = \mathrm{E}_0 \mathrm{A}_0 [\varepsilon_\mathrm{I}(t) - \varepsilon_\mathrm{R}(t)] = \mathrm{E}_0 \mathrm{A}_0 \varepsilon_\mathrm{T}(t) \\ \varepsilon(t) = \frac{2\mathrm{C}_0}{\mathrm{D}} \int_0^\tau [\varepsilon_\mathrm{I}(t) - \varepsilon_\mathrm{R}(t)] dt = \frac{2\mathrm{C}_0}{\mathrm{D}} \int_0^\tau \varepsilon_\mathrm{R}(t) dt \\ \dot{\varepsilon}(t) = -\frac{2\mathrm{C}_0}{\mathrm{D}} [\varepsilon_\mathrm{I}(t) - \varepsilon_\mathrm{T}(t)] = -\frac{2\mathrm{C}_0}{\mathrm{D}} \varepsilon_\mathrm{R}(t) \end{array}\right\} \tag{4}$$

where: $\varepsilon_I$, $\varepsilon_R$ and $\varepsilon_T$ are the incident, reflected, and transmitted strains; $\tau$ is the duration of stress wave; $C_0$ is the longitudinal wave velocity of the pres rod $C_0 = \sqrt{E_0/\rho_0}$; $A_0$ is the cross-sectional area of the press rod

### 3.2. Dynamic Performance Analysis

The dynamic performance parameters of ring sandstone specimens with and without temperature and water coupling are shown in Table 2.

**Table 2.** SHPB test data.

| Inner Diameter (mm) | Specimen Number | After Temperature and Water Coupling Treatment | | | | Specimen Number | Without Temperature and Water Coupling Treatment | | | |
|---|---|---|---|---|---|---|---|---|---|---|
| | | Peak Stress (MPa) | Average Strain Rate (s$^{-1}$) | Dynamic Modulus of Elasticity (GPa) | Peak Strain ($\times 10^{-3}$) | | Peak Stress (MPa) | Average Strain Rate (s$^{-1}$) | Dynamic Modulus of Elasticity (GPa) | Peak Strain ($\times 10^{-3}$) |
| 0 (Intact) | DJ04-01 | 123.46 | 8.40 | 13.07 | 94.70 | DJ04-04 | 145.70 | 7.62 | 14.41 | 87.60 |
| | DJ04-03 | 122.20 | 8.60 | 13.04 | 92.30 | DJ04-06 | 143.85 | 7.93 | 15.01 | 81.00 |
| 5 | DJ04-07 | 123.46 | 9.62 | 14.30 | 91.10 | DJ04-11 | 126.26 | 8.83 | 14.45 | 89.70 |
| | DJ04-08 | 122.20 | 8.93 | 10.67 | 97.80 | DJ04-12 | 131.77 | 8.62 | 13.88 | 91.80 |
| 10 | DJ04-13 | 117.10 | 11.45 | 12.36 | 96.60 | DJ04-17 | 127.04 | 10.77 | 13.49 | 95.30 |
| | DJ04-15 | 120.95 | 11.35 | 12.24 | 100.60 | DJ04-18 | 120.50 | 11.18 | 13.51 | 94.70 |
| 15 | DJ04-19 | 107.42 | 14.81 | 10.11 | 126.40 | DJ04-22 | 135.24 | 11.48 | 12.76 | 106.50 |
| | DJ04-20 | 111.10 | 12.12 | 10.82 | 118.40 | DJ04-23 | 103.17 | 11.36 | 10.11 | 111.60 |
| 20 | DJ04-25 | 69.66 | 15.72 | 8.80 | 161.70 | DJ04-28 | 112.13 | 11.73 | 9.65 | 129.00 |
| | DJ04-26 | 79.39 | 15.93 | 7.39 | 146.20 | DJ04-30 | 89.00 | 15.30 | 9.12 | 139.40 |
| 25 | DJ04-31 | 60.44 | 18.85 | 8.03 | 173.20 | DJ04-34 | 89.00 | 15.30 | 8.12 | 158.40 |
| | DJ04-33 | 59.80 | 18.55 | 7.17 | 163.70 | DJ04-35 | 51.30 | 16.93 | 9.00 | 157.70 |

### 3.2.1. Dynamic Stress-Strain Relationship

After analyzing and processing the two-wave signals collected by the strain gauge, the dynamic stress-strain relationship diagram of the ring sandstone specimen was obtained. Figure 8. describes the dynamic stress-strain relationship.

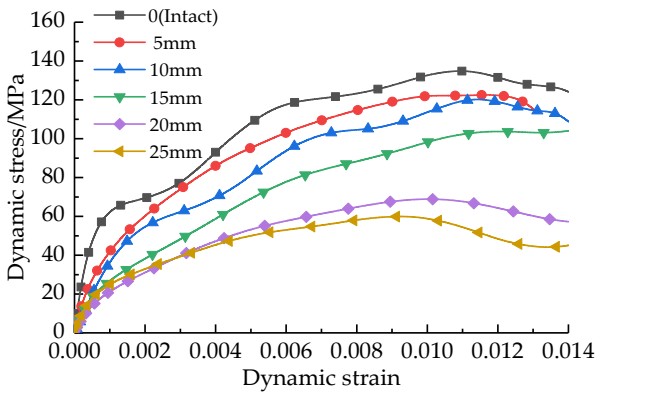

(**a**) After Temperature and Water Coupling Treatment

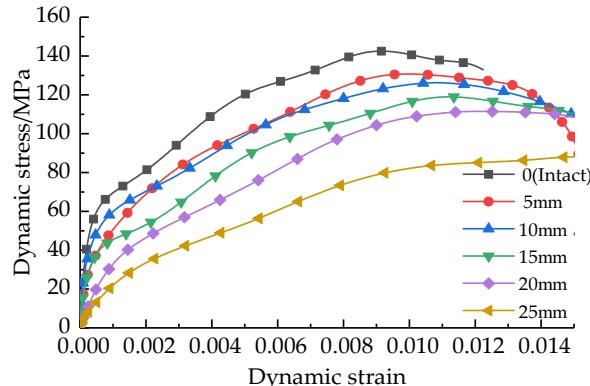

(**b**) Without Temperature and Water Coupling Treatment

**Figure 8.** Dynamic stress-strain curve of sandstone specimen.

The slope of the dynamic stress-strain curve of sandstone specimens with and without temperature and water coupling ring is gradually gentle with the increase in inner diameter, which is shown in Figure 9. The intact sandstone specimen has the biggest dynamic compressive strength and the lowest dynamic peak strain. The dynamic compressive strength of the ring sandstone specimen is smaller, and the dynamic peak strain is larger after temperature and water coupling.

Analysis of the reasons, the coupling effect of temperature and water caused deterioration damage to the ring sandstone specimen, and the dynamic performance of the specimen was weakened.

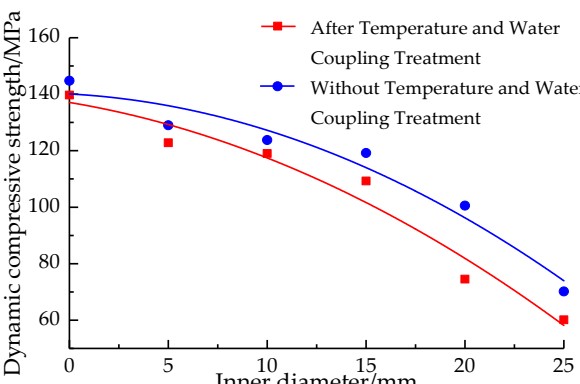

**Figure 9.** The relationship between dynamic compressive strength and annular diameter change.

### 3.2.2. Dynamics Performance

The Figure 9 is about the relationship between dynamic compressive strength with inner diameter.

The dynamic compressive strength after temperature and water coupling are lower than that without temperature and water coupling. The dynamic compressive strength of specimens with and without temperature and water coupling reaches the maximum value in the intact state. After temperature and water, the maximum dynamic compressive strength of the ring sandstone specimen coupling is 139.71 MPa, and the maximum dynamic compressive strength of the ring sandstone specimen without temperature and water coupling is 144.78 MPa. It shows a quadratic function relationship that decreases with the increase in inner diameter, and the negative correlation is obvious. The fitting formula is shown in (5). The correlation coefficients reached 0.9406 and 0.9303, respectively, showing a strong correlation.

$$\left.\begin{array}{l} \sigma_0(d) = -0.080d^2 - 1.170d + 137.091 \quad (\mathrm{R}^2 = 0.9406) \\ \sigma_\mathrm{T}(d) = -0.091d^2 - 0.381d + 140.103 \quad (\mathrm{R}^2 = 0.9303) \end{array}\right\} \tag{5}$$

where, $\sigma_0(d)$ and $\sigma_\mathrm{T}(d)$ are respectively, the dynamic compressive strength of specimens treated with temperature and water coupling and those treated without warm water coupling.

Analysis of the reasons, the internal crack of the specimen expands with a tendency of expansion. The wall thickness of ring sandstone specimens decreases with the increase in inner diameter, the resistance to deformation weakens, and the dynamic compressive strength shows a decreasing trend.

### 3.2.3. Dynamic Peak Strain

Figure 10 shows the dynamic peak strain and inner diameter.

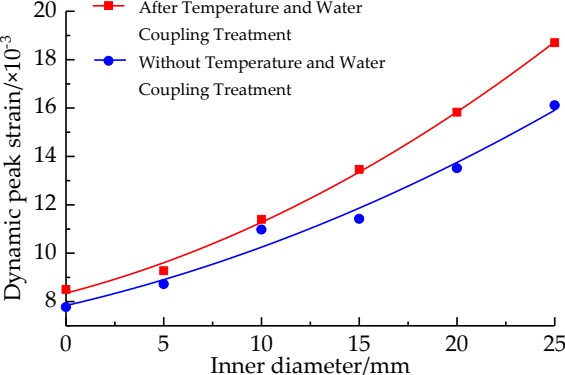

**Figure 10.** Diagram of the dynamic peak strain and annular diameter change.

The specimen's dynamic peak strain which treated with temperature and water coupling, is greater than without treatment. The dynamic peak strain of both specimens reached its maximum value at the inner diameter of 25 mm, which were $18.55 \times 10^{-3}$ and $16.93 \times 10^{-3}$, respectively. The fitting analysis was carried out, and the correlation coefficient reached 0.9965 and 0.9696. The fitting formula was obtained as shown in (6)

$$\left.\begin{aligned} \varepsilon_0(d) &= 0.008d^2 + 0.211d + 8.348 \quad (R^2 = 0.9965) \\ \varepsilon_T(d) &= 0.005d^2 + 0.187d + 7.838 \quad (R^2 = 0.9696) \end{aligned}\right\} \tag{6}$$

where, $\varepsilon_0$ and $\varepsilon_T$ are respectively, the dynamic peak strain of specimens with and without temperature and water coupling treatment.

It is found that the dynamic peak strain of ring sandstone specimens is related to the dynamic compressive strength of ring sandstone specimens. The greater the dynamic compressive strength of the specimen, the smaller the degree of breakage and the smaller the dynamic peak strain. Therefore, the coupling effect of temperature and water affects the dynamic properties of the specimen from the aspect of internal structural damage.

### 3.2.4. Average Strain Rate

The variation of average strain rate with specimen inner diameter is shown in Figure 11.

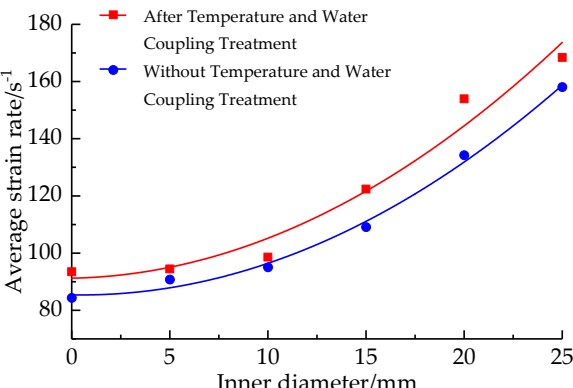

**Figure 11.** Diagram of average strain rate and annular diameter change.

The average strain rate of ring sandstone specimens treated with and without temperature and water coupling showed an increasing trend with the increase in inner diameter. The fitting formula is shown in (7). The correlation coefficients reached 0.9478 and 0.9912. The average strain rate reaches its maximum value at the inner diameter of 25 mm. The average strain rate of ring sandstone specimen under temperature and water coupling is $168.45 \text{ s}^{-1}$, and that without temperature and water is $158.05 \text{ s}^{-1}$.

$$\left.\begin{aligned} \dot{\varepsilon}_0(d) &= 0.127d^2 + 1.130d + 91.230 \quad (R^2 = 0.9478) \\ \dot{\varepsilon}_T(d) &= 0.122d^2 - 0.114d + 85.392 \quad (R^2 = 0.9912) \end{aligned}\right\} \tag{7}$$

where, $\dot{\varepsilon}_0$ and $\dot{\varepsilon}_T$ are respectively the average strain rates of specimens with and without temperature and water coupling.

Analysis of the reasons, the coupling effect of temperature and water leads to the ring specimen's dynamic compressive strength decreasing, and peak strain increasing, so the specimen is more prone to failure. As the inner diameter of the ring sandstone specimen increases, the deformation resistance of the specimen is weakened.

### 3.2.5. Dynamic Modulus of Elasticity

The dynamic elastic modulus and inner diameter is shown in Figure 12.

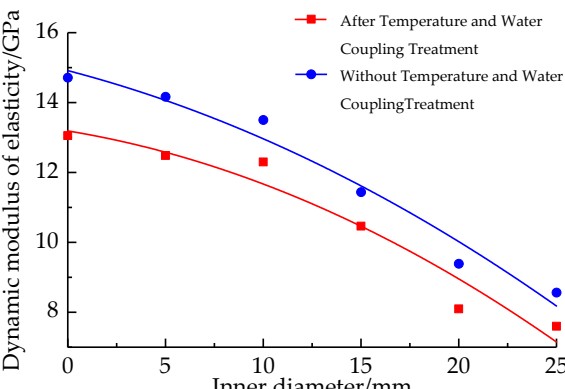

**Figure 12.** Relationship between dynamic elastic modulus and annular diameter change.

The fitting formula is shown in Formula (8). The correlation coefficients were 0.9178 and 0.9537. The dynamic elastic moduli of intact sandstone specimens are the largest, which are 7.60 GPa and 8.56 GPa.

$$\left.\begin{array}{l} E_0(d) = -0.006d^2 - 0.092d + 13.189 \quad (\text{R}^2 = 0.9178) \\ E_\text{T}(d) = -0.005d^2 - 0.146d + 14.914 \quad (\text{R}^2 = 0.9537) \end{array}\right\} \tag{8}$$

where, $E_0(d)$ and $E_\text{T}(d)$ are, respectively, the dynamic modulus of elasticity with temperature and water coupling and without temperature and water coupling.

The analysis shows that the wall thickness of the specimen decreases with the increase in the inner diameter, which makes it more prone to instability failure. The dynamic elastic modulus is reduced. The specimen shows the characteristic of decreasing elastic modulus after the coupling action of temperature and water. Because the coupling action causes deterioration damage in the sandstone specimen

## 4. Analysis of Fracture Morphology and Energy Dissipation of Specimen

### 4.1. Fracture Surface

The ring sandstone specimens were placed into (Scanning Electron Microscopy, SEM) for observation of the microscopic fracture surface. The SEM photos of the sandstone ring specimen are shown in Figure 13.

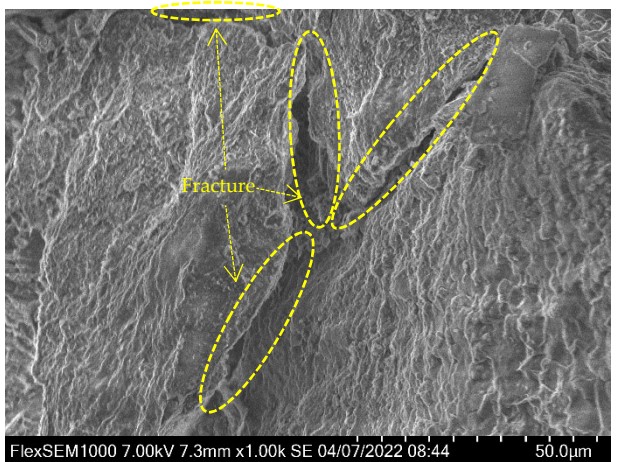
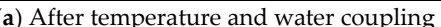

(**a**) After temperature and water coupling

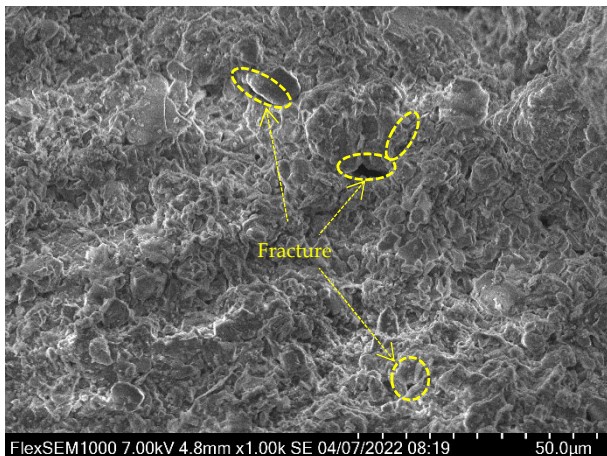

(**b**) Without temperature and water coupling

**Figure 13.** SEM test photos of sandstone specimens.

As can be seen from Figure 13, cracks in ring sandstone specimens after temperature and water coupling are wider than those without temperature and water coupling and have a tendency to expand.

After analyzing the reasons, the inner cracks of the ring sandstone specimen expanded and expanded, and the stability of the specimen decreased. Therefore, the sandstone specimen with temperature and water coupling is more likely to be damaged than the sandstone ring specimen without temperature and water coupling, and its dynamic performance is weaker than that without temperature and water coupling ring sandstone specimen.

### 4.2. SHPB Test Facility

Observe and analyzing the fracture morphology of the specimen, which has been shown in Figures 14 and 15.

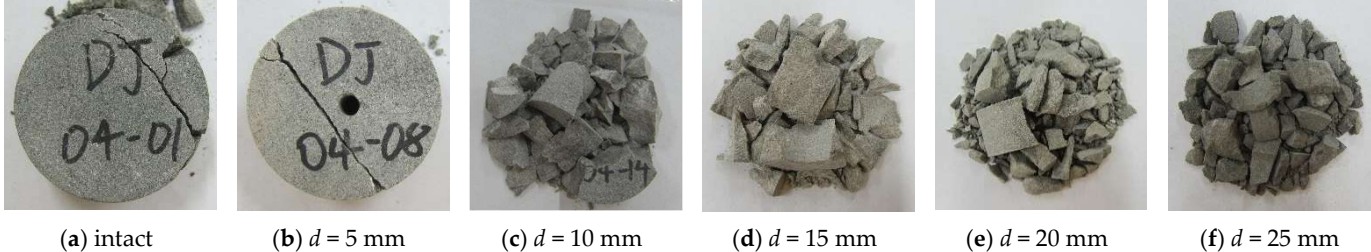

| (**a**) intact | (**b**) *d* = 5 mm | (**c**) *d* = 10 mm | (**d**) *d* = 15 mm | (**e**) *d* = 20 mm | (**f**) *d* = 25 mm |

**Figure 14.** Fracture morphology of specimens with temperature and water coupling.

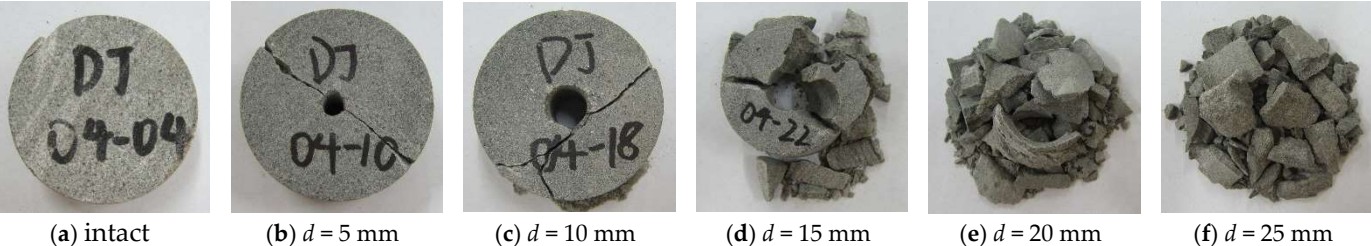

| (**a**) intact | (**b**) *d* = 5 mm | (**c**) *d* = 10 mm | (**d**) *d* = 15 mm | (**e**) *d* = 20 mm | (**f**) *d* = 25 mm |

**Figure 15.** Fracture morphology of specimens without temperature and water coupling.

It can be seen from Figure 15 that the integrity of sandstone specimens decreases with the increase in the specimen's inner diameter. The shape integrity of the ring sandstone specimen with warm water coupling is obviously less than that without warm water coupling. The fracture degree of the ring sandstone specimen increases with the increase in the specimen's inner diameter.

After coupling with temperature and water, the inner cracks of the ring sandstone specimen were damaged by expansion. When the progressive load is loaded, the ring sandstone specimen first cracks along the internal cracks of the specimen, and then several cracks extend from the initiation parts, leading to the failure of the ring sandstone specimen. As the inner diameter of the ring sandstone specimen increases, the inner wall of the specimen decreases relatively, and the mutual restraint of the cracks in the specimen will be weakened, so the specimen is more prone to instability and failure.

### 4.3. Energy Dissipation Analysis

The energy formula is shown in (9).

$$\left.\begin{array}{l} W_I(t) = E_0 C_0 A_0 \int_0^t \varepsilon_I^2(t)dt \\ W_R(t) = E_0 C_0 A_0 \int_0^t \varepsilon_R^2(t)dt \\ W_T(t) = E_0 C_0 A_0 \int_0^t \varepsilon_T^2(t)dt \\ W_D(t) = W_I(t) - W_R(t) - W_T(t) \end{array}\right\} \tag{9}$$

where, $W_I(t)$, $W_R(t)$, $W_T(t)$, and $W_D(t)$ are, respectively, incident energy, reflected energy, transmitted energy, and absorbed energy.

The energy data of the specimen is shown in Table 3 after processing.

**Table 3.** Energy data of sandstone specimen.

| Inner Diameter (mm) | Specimen Number | After Temperature and Water Coupling Treatment | | | | Specimen Number | Without Temperature and Water Coupling Treatment | | | |
|---|---|---|---|---|---|---|---|---|---|---|
| | | Incident Energy (J) | Reflected Energy (J) | Transmitted Energy (J) | Absorbed Energy (J) | | Incident Energy (J) | Reflected Energy (J) | Transmitted Energy (J) | Absorbed Energy (J) |
| 0 (Intact) | DJ04-01 | 104.46 | 29.66 | 53.22 | 21.58 | DJ04-04 | 117.76 | 39.33 | 60.37 | 18.06 |
| | DJ04-03 | 100.36 | 28.75 | 51.32 | 20.29 | DJ04-06 | 114.44 | 33.31 | 62.35 | 18.78 |
| 5 | DJ04-07 | 95.77 | 28.92 | 45.12 | 21.74 | DJ04-11 | 113.82 | 36.11 | 58.69 | 19.02 |
| | DJ04-08 | 107.12 | 30.92 | 55.69 | 20.58 | DJ04-12 | 113.04 | 36.17 | 57.94 | 18.92 |
| 10 | DJ04-13 | 106.00 | 31.21 | 52.05 | 21.74 | DJ04-17 | 112.51 | 38.16 | 54.61 | 19.74 |
| | DJ04-15 | 103.53 | 34.48 | 46.67 | 22.38 | DJ04-18 | 110.81 | 35.06 | 55.86 | 19.89 |
| 15 | DJ04-19 | 107.32 | 38.08 | 46.62 | 22.62 | DJ04-22 | 113.63 | 41.19 | 52.28 | 20.16 |
| | DJ04-20 | 105.70 | 37.74 | 45.03 | 22.93 | DJ04-23 | 114.52 | 43.07 | 50.73 | 20.72 |
| 20 | DJ04-25 | 104.95 | 43.25 | 38.06 | 23.64 | DJ04-28 | 115.42 | 45.96 | 47.65 | 21.82 |
| | DJ04-26 | 101.19 | 43.45 | 34.66 | 23.08 | DJ04-30 | 113.71 | 45.35 | 45.94 | 22.42 |
| 25 | DJ04-31 | 105.68 | 51.58 | 29.50 | 24.60 | DJ04-34 | 112.66 | 49.02 | 40.98 | 22.66 |
| | DJ04-33 | 105.90 | 51.12 | 30.61 | 24.17 | DJ04-35 | 112.00 | 47.93 | 40.72 | 23.36 |

As the inner diameter of the ring sandstone specimen increases, the reflected energy and absorbed energy of the specimen have an increasing trend, while the transmitted energy shows a decreasing trend. Under the same loading pressure, the incident energy basically remains unchanged, so the energy ratio can be used to study the energy dissipation law of specimens.

Definition $C_R' = W_R/W_I$ (reflected energy/incident energy); $C_T' = W_T/W_I$ (transmitted/incident energy); $C_D' = W_D/W_I$ (Absorbed energy/incident energy). The energy proportion diagram is shown in Figure 16.

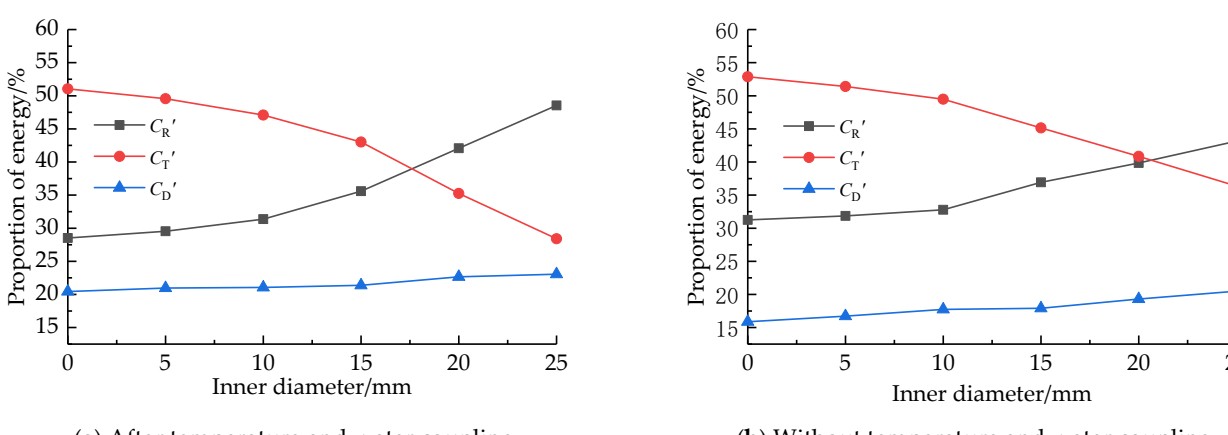

(**a**) After temperature and water coupling

(**b**) Without temperature and water coupling

**Figure 16.** Energy proportion diagram.

The proportion of absorbed energy of the ring sandstone specimen with temperature and water coupling is greater than that without temperature and water coupling. The absorption energy ratio of the ring sandstone specimen increases gradually with the increase in inner diameter.

Because the inner diameter of the ring sandstone specimen is different, the volume of the specimen will change. The energy dissipation ratio per unit volume was defined (energy dissipation per unit volume), and the energy dissipation analysis was further carried out on the axial compression failure of the ring specimen. Draw the energy dissipation diagram per unit volume as shown in Figure 17.

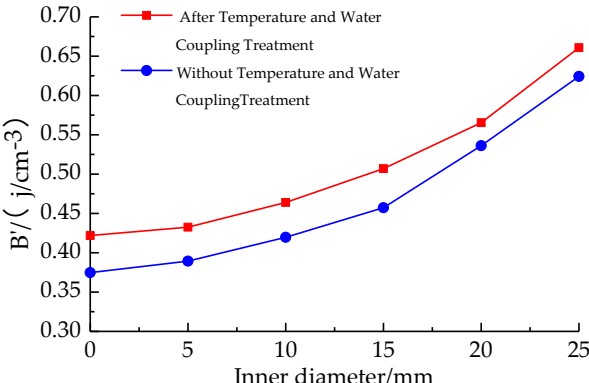

**Figure 17.** Energy dissipation per unit volume diagram.

The energy dissipation per unit volume of the specimen increases can be seen in Figure 17. The energy dissipation per unit volume can reflect the broken shape of the specimen.

According to the analysis of reasons, the energy absorbed by the specimen is used to penetrate the cracks of the specimen, resulting in compression failure of the specimen. Therefore, the greater the degree of breakage of the specimen, the more energy dissipation per unit volume of the specimen. With the increase in inner diameter, energy dissipation per unit volume of the specimen increases, so the degree of breakage increases. The absorption energy dissipation per unit volume of sandstone specimen with temperature and water coupling ring is greater than that without temperature and water coupling ring sandstone specimen. Therefore, the breakage degree of the sandstone specimen with temperature and water coupling ring is greater.

## 5. Conclusions

The dynamic mechanical properties of ring sandstone specimens with different inner diameters after and without temperature and water coupling were studied to provide certain data support for the analysis of coal mine roadway safety support under the combined influence of underground water and high ground temperature. After analysis, the conclusion is as follows.

(1) After coupling with temperature and water, the mass growth rate of ring sandstone specimen is about 0.2%; the Density and volume growth rate is about 0.6%. The density showed a decreasing trend with a reduction rate of 0.004%. XRD results show that the material composition of ring sandstone samples does not change under the coupling effect of temperature and water, and no new materials are found.

(2) The dynamic properties of ring sandstone specimens decrease with the increase inin inner diameter. The dynamic properties of the ring sandstone specimen were weakened by temperature and water coupling.

(3) Both dynamic compressive strength and the peak strain show a quadratic function relationship with the increase in inner diameter, and the positive correlation is obvious. The average strain rate and dynamic elastic modulus show a strong quadratic negative correlation with the increase in specimen inner diameter.

(4) Through the microscopic fracture surface analysis, the coupling effect of temperature and water damaged the ring sandstone specimen through degradation, which made the specimen rupture and expansion, and more prone to ring breakage. The crack of the specimen tends to expand, which increases the degree of breakage of the specimen.

(5) Through the energy dissipation analysis, the energy dissipation per unit volume of the ring sandstone specimen increases after the coupling of warm water. The law of energy dissipation is consistent with the broken form of the specimen.

**Author Contributions:** Conceptualization, Q.P.; Software, Q.P.; Validation, Q.G.; Formal analysis, Q.G.; Data curation, Q.G.; Writing—original draft, Q.G.; Writing—review & editing, S.W. All authors have read and agreed to the published version of the manuscript.

**Funding:** National Natural Science Foundation of China: no. 52074005; National Natural Science Foundation of China: no. 52074006; Anhui University of Science and Technology Graduate Innovation Fund Project: no. 2021CX2032; National College Student Innovation and Entrepreneurship Training Program: no. 202110361022, no. 2021103661027, no. 202110361032.

**Data Availability Statement:** The data used to support the findings of this study are available from the corresponding author upon request.

**Conflicts of Interest:** The authors declare no conflict of interest. The funders had no role in the design of the study; in the collection, analyses, or interpretation of data; in the writing of the manuscript, or in the decision to publish the results.

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
