# Peer review of "Study on Mechanical Properties of Ring Sandstone Specimen under Temperature and Water Coupling Dynamic Compression"

_minerals, doi:10.3390/min13010119_

Round 1

Reviewer 1 Report

Please find the detailed comments in the attached file.

Reviewer 2 Report

The content and idea of the article are novel, and the research content has certain significance for engineering construction. It is suggested that the article can be accepted after minor revision. For this, I have the following questions and suggestions:

1The part of the abstract needs to be expanded appropriately, and some research results are not given.

2The introduction part of the paper needs to be refined, which can appropriately increase the significance of this experiment for engineering research.

3The fourth section of the article "broken form" should appropriately add some descriptions

4The conclusion suggests reorganizing the language and rearranging it.

Reviewer 3 Report

The topic is experimental studies of fracture in sandstone under the influence of water saturation and temperature, in the specific context of coal mine roadway stability. The topic is interesting, although it seems like a stretch to see this material in a journal called Minerals.

There are three areas where the manuscript can be improved. (1)  The material tested is described as ‘sandstone’ but this is pretty vague given the wide range of compositions, grain sizes, porosities, and cement contents of sandstones. These differences in composition have been shown to influence the subcritical (chemically and fluid assisted) propagation of fractures (e.g. Reviews of Geophysics, 2019, 57 (3), 1065-1111. doi:10.1029/2019RG000671); consequently, it would improve the MS to provide more information about the characteristics of the sandstone. The reference list of the Reviews of Geophysics paper—open access—could be helpful in this regard. (2) The text needs some further polishing for usage. See the example below. (3) The referencing seems parochial to me. Coal mine and other subterranean excavation stability is an old field with a lot of previous work; mentioning a wider swath of the literature and situating your new work relative to other recent studies seems like it should be in order; additionally, the way alerting services work these days, your paper is more likely to be noticed in the international literature if you call out some of this work.

In the first line of the Introduction; suggested revision: “Coal mine roadways created during tunneling are complex temperature, fluid saturation, and loading environments posing great challenges in roadway excavation and support”. This revision fixes some grammatical issues, and more specifically alerts readers to the topics you will cover.
